# Spurious early ecological association suggesting BCG vaccination effectiveness for COVID-19

Jorge R. Ledesma[1], Peter Lurie[2], Rachel R. Yorlets[1,3], Garrison Daly[2], Stavroula Chrysanthopoulou[4], Mark N. Lurie [1,3]*

**1** Department of Epidemiology, Brown University School of Public Health, Providence, Rhode Island, United States of America, **2** Center for Science in the Public Interest, Washington, D.C., United States of America, **3** Population Studies and Training Center, Brown University, Providence, Rhode Island, United States of America, **4** Department of Biostatistics, Brown University School of Public Health, Providence, Rhode Island, United States of America

* Mark_Lurie@brown.edu

## Abstract

### Background

Several ecologic studies have suggested that the bacillus Calmette-Guérin (BCG) vaccine may be protective against SARS-CoV-2 infection including a highly-cited published pre-print by Miller *et al.*, finding that middle/high- and high-income countries that never had a universal BCG policy experienced higher COVID-19 burden compared to countries that currently have universal BCG vaccination policies. We provide a case study of the limitations of ecologic analyses by evaluating whether these early ecologic findings persisted as the pandemic progressed.

### Methods

Similar to Miller *et al.*, we employed Wilcoxon Rank Sum Tests to compare population medians in COVID-19 mortality, incidence, and mortality-to-incidence ratio between countries with universal BCG policies compared to those that never had such policies. We then computed Pearson's r correlations to evaluate the association between year of BCG vaccination policy implementation and COVID-19 outcomes. We repeated these analyses for every month in 2020 subsequent to Miller *et al.*'s March 2020 analysis.

### Results

We found that the differences in COVID-19 burden associated with BCG vaccination policies in March 2020 generally diminished in magnitude and usually lost statistical significance as the pandemic progressed. While six of nine analyses were statistically significant in March, only two were significant by the end of 2020.

**Data Availability Statement:** The data underlying the results presented in the study are available from the COVID-19 Data Repository by the Center for Systems Science and Engineering (CSSE) at

Johns Hopkins University (https://github.com/
CSSEGISandData/COVID-19).

**Funding:** ML and PL receive funding from the
Sydney E. Frank Foundation. The funder had no
role in study design, data collection and analysis,
decision to publish, or preparation of the
manuscript.

**Competing interests:** The authors have declared
that no competing interests exist.

## Discussion

These results underscore the need for caution in interpreting ecologic studies, given their
inherent methodological limitations, which can be magnified in the context of a rapidly evolv-
ing pandemic in which there is measurement error of both exposure and outcome status.

## Introduction

Since the identification of SARS-CoV-2 in December 2019 in Wuhan, China, COVID-19 has
spread to humans in every country in the world, resulting in over 250 million infections and
five million deaths as of this writing [1]. Throughout the pandemic, scientists have considered
the possibility of repurposing existing preventive and therapeutic agents, including the cen-
tury-old [2] bacillus Calmette-Guérin (BCG) vaccine. Currently, 157 countries mandate uni-
versal BCG vaccination, and 19 countries (*e.g.*, the United States, Italy, Belgium) only
vaccinate high-risk groups [3]. In 2020, laboratory analyses found that BCG vaccination
induces specific immunity against the envelope protein of SARS CoV-2 [4].

However, to date, most of the evidence for BCG protection against COVID-19 has come
from a series of ecologic studies, including a highly-cited pre-print by Miller and colleagues in
March 2020 [5]. In that paper, Miller *et al.* found that middle/high- and high-income countries
that never had a universal BCG policy experienced higher COVID-19 incidence and mortality
compared to countries that currently have universal BCG vaccination policies. In addition,
this analysis suggested that the older the universal BCG policy (and thus presumably the larger
the proportion of the population vaccinated), the lower the national COVID-19 mortality rate.
Since this early work, there have been several observational studies and a meta-analysis of four
ecologic studies that yielded a statistically significant pooled negative correlation between the
percentage of the population vaccinated with BCG and COVID-19 mortality at the country
level (random effect pooled r: -0.48 [95% confidence interval: -0.61 to -0.35]) [6].

In April 2020, the WHO announced that, in the absence of positive results from RCTs (the
gold standard for evaluating vaccine efficacy), there was insufficient evidence of a causal rela-
tionship between the BCG vaccine and COVID-19 infection [7]. As of March 2021, there were
more than two dozen clinical trials underway across the globe to evaluate whether BCG pro-
tects against COVID-19. Completed assessments to date include a case-control study [8] that
did not demonstrate effectiveness and an unpublished RCT [9] conducted in elderly patients
in Greece claiming effectiveness that is limited by small sample size and high loss to follow-up.
In the interim, we aimed to evaluate whether the early ecologic findings that BCG appeared to
protect against COVID-19 persisted as the pandemic progressed.

## Methods

To assess the robustness of the Miller *et al.* findings, we replicated their analysis using identical
statistical methods for each month in 2020 subsequent to their March 2020 analysis. Those
authors examined the relationship between BCG and COVID-19 morbidity and mortality
from January 1, 2020 –March 21, 2020 using COVID-19 data from Google News. Because
Google News does not provide historical data by month, we utilised data on COVID-19 inci-
dence and mortality from the COVID-19 Data Repository at Johns Hopkins University. This
database compiles data from several reliable sources, including the World Health Organiza-
tion, several country-specific Centers for Disease Control and Prevention and ministries of
health, and provides historical data by month [1].

A comparison between the Miller *et al.* March 2020 dataset and the March 2020 dataset generated by the Johns Hopkins Repository showed these two sources yield similar results, with the Hopkins Repository providing on average 1.07 [range: 0.00 to 2.45] and 1.09 [range: 0.50 to 2.33] times higher mortality and incidence counts, respectively, for each country. In addition, replicating the Miller *et al.* analyses with the Hopkins Repository data for March provided similar statistical results at the p = 0.05 level.

Like Miller *et al.*, we evaluated three outcomes: per capita COVID-19 mortality (number of deaths from COVID-19 per 100,000 population), incidence (number of COVID-19 cases per 100,000 population), and mortality-to-incidence ratio (the number of deaths divided by the number of cases). Like Miller *et al.*, we extracted data on country-specific BCG vaccination policies from the BCG World Atlas [3], restricting our analyses to countries that 1) have a population of over one million people, and 2) are classified as middle/high- or high-income per World Bank categories [10]. Wilcoxon Rank Sum Tests were then used to compare population medians between countries with universal BCG policies and those that never had such policies. Pearson's r correlations were also computed to evaluate the association between year of BCG vaccination policy implementation and COVID-19 outcomes. Statistical tests were considered significant if p<0.05 (2-sided).

## Results

### Relationship between policies and outcomes

Miller *et al.* examined 55 middle/high- and high-income countries that had universal BCG polices in place and five that had never had a universal BCG vaccination policy. In March 2020, those countries with universal vaccination policies had statistically significantly lower COVID-19 mortality, incidence, and mortality-to-incidence ratios than those that never had such BCG policies using Wilcoxon Rank Sum Tests. When we repeated their analyses for subsequent months, we initially found that COVID-19 mortality rates in countries with no history of BCG policies continued to significantly exceed those in counties with current BCG policies, with the magnitude of that difference essentially stable after May 2020 (Fig 1). In contrast, differences between these groups with respect to COVID-19 incidence were not significant from May to October, although they became statistically significant again in November and December. For COVID-19 mortality-to-incidence ratios, statistically significant differences were present through July, but not thereafter.

### Relationship between duration of policies and outcomes

Miller *et al.* also examined the correlation between the number of years of implementation of the BCG policy (for the 45 countries for which data on year of policy implementation were available) and the same three COVID-19 outcomes. Twenty-eight countries had current universal BCG policies, while 17 previously had them.

For those with current BCG policies (Fig 2A), a statistically significant positive association was present for mortality rate in March 2020 but not thereafter; Pearson correlation coefficients for incidence rate and mortality-to-incidence ratio were not statistically significant at any period but the correlation coefficients were lower in all months than in March 2020 (and sometimes negative suggesting that more recent BCG policy implementation was associated with lower burden of COVID-19). For countries with previous universal BCG vaccination policies (Fig 2B), statistically significant findings for mortality and incidence in March 2020 were no longer apparent by April and May 2020, respectively, and remained non-significant throughout the study period. The correlation coefficient for mortality-to-incidence ratio was never statistically significant. Except for a small increase in the correlation coefficient for

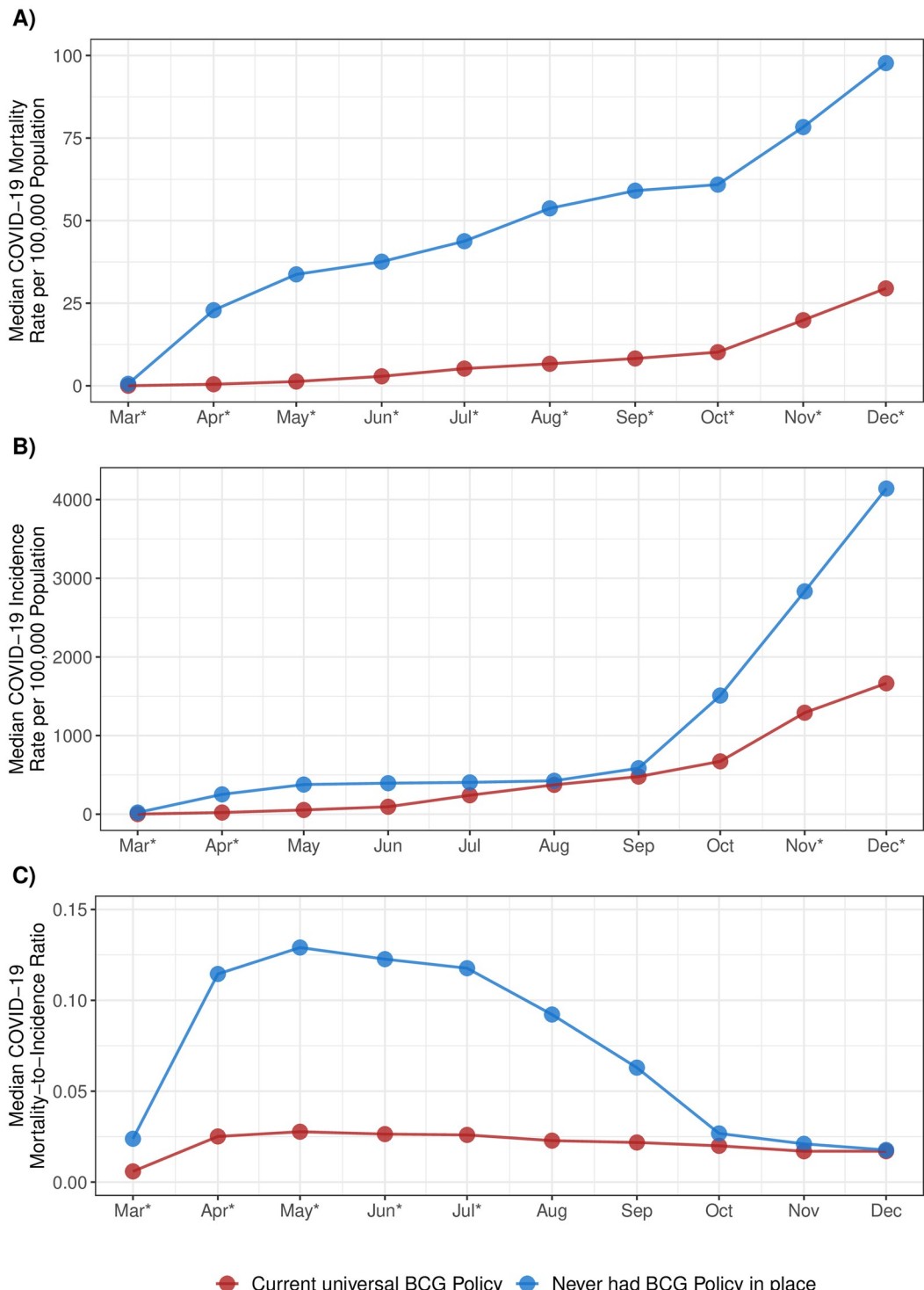

**Fig 1. Median COVID-19 mortality rate per 100,000 population (A), incidence rate per 100,000 population (B), and mortality-to-incidence ratio (C) by BCG policy status, March-December, 2020.** * Statistically significant at p<0.05 using Wilcoxon Rank Sum Test.

**A) countries with current universal BCG vaccination policy**

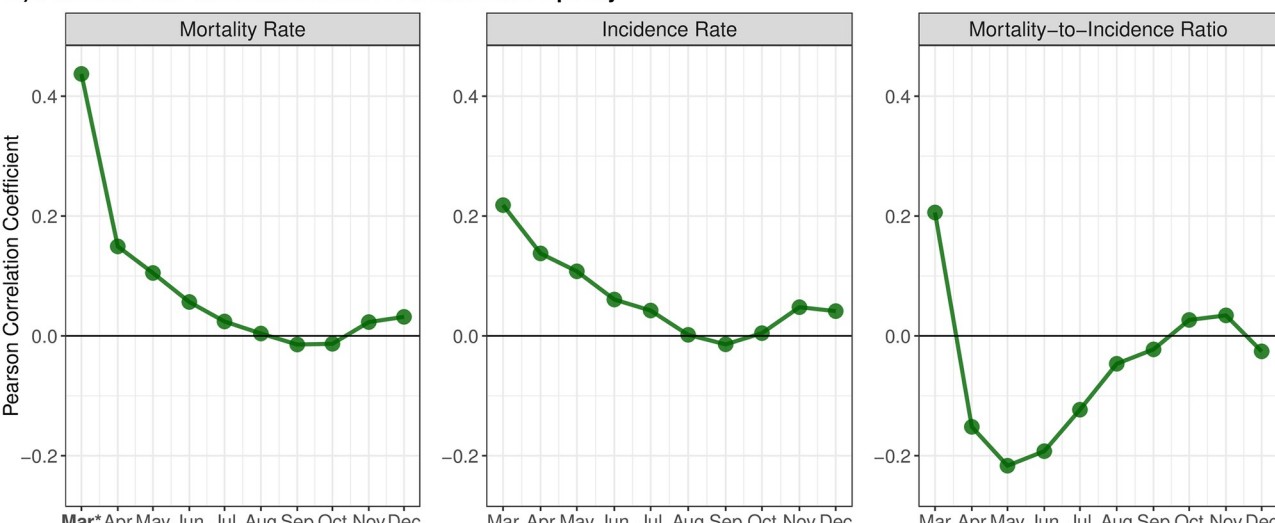

**B) countries with past universal BCG vaccination policy**

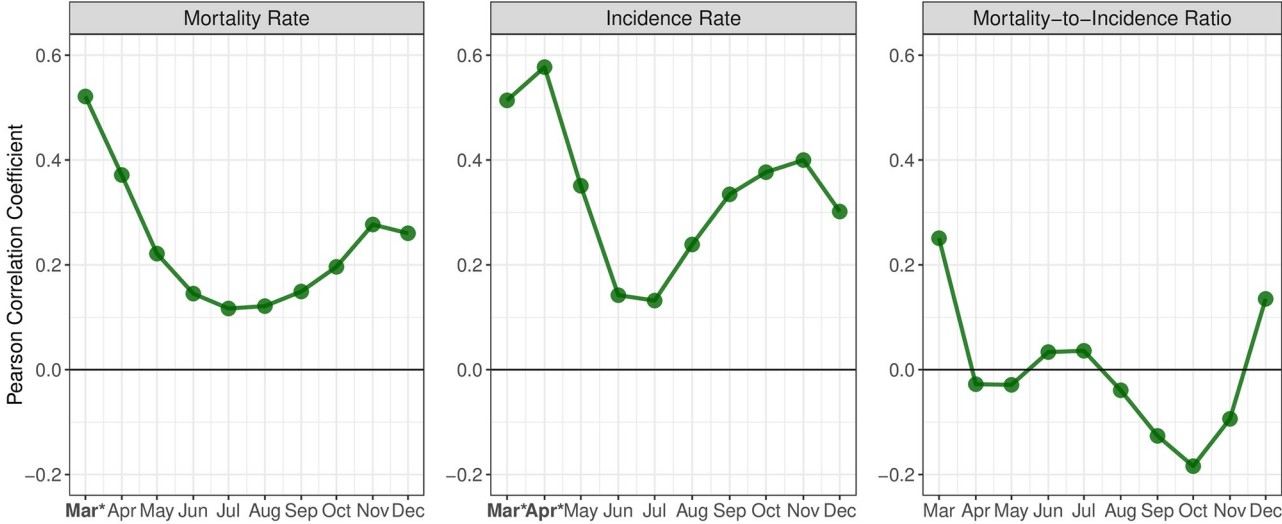

**Fig 2. Pearson correlation coefficients between years of implementation of BCG Policy and COVID-19 outcomes among countries with current universal BCG vaccination policy (A) and among countries that previously had universal BCG vaccination policies (B), March-December, 2020.** [*] Statistically significant Pearson correlation at p<0.05.

incidence in April 2020 among countries that no longer have universal BCG policies, all correlation coefficients were lower in every month after March 2020 for all three outcomes.

## Discussion

Our analysis demonstrates that the differences in COVID-19 burden associated with BCG vaccination policies that Miller *et al.* observed in March 2020 generally diminished in magnitude and usually lost statistical significance as the pandemic progressed. While six of nine analyses were statistically significant in March, only two were significant by the end of 2020, including none of the six correlations between year of BCG policy implementation and COVID-19 burden, three of which were originally considered statistically significant.

Our case study illustrates that the timing of the initial analysis was an important factor in the apparent detection of associations, as we found that the indices measured by Miller *et al.* varied substantially and typically diminished in magnitude over time. This finding was also observed in a more limited confirmatory analysis, in which Lindestam Arlehamn *et al.* repeated the analysis of a different ecologic study [11] and found that certain associations apparent in March 2020 were no longer present in August 2020 [12].

Ecologic studies have provided valuable initial understandings of the effects of various exposures on population-level disease outcomes. Innovative ecologic studies of lactation on breast cancer [13], salt intake and blood pressure [14], and male circumcision on HIV transmission [15], have provided important insights into causal hypotheses at the individual-level. Other advantages of these studies includes the ability to efficiently examine trends over time, similar to the analysis here where the BCG-COVID relationships were reassessed as the pandemic progressed. Further, ecologic studies are a critical part of the methodological toolbox for evaluating the impact of policy changes and interventions by examining pre and post population-level outcomes.

However, the limitations of ecologic studies have also been well-described and many are magnified here. First, the ecologic fallacy describes the hazards of using population-level data to make inferences at the individual level [16]. Second, neither dichotomising countries by national BCG vaccination policies nor the number of years policies were in effect captures the substantial variability in actual vaccination coverage [3, 17]. Third, particularly in the early phases of the pandemic, there was wide variation in testing rates, case definitions, and case reporting. Recent modeling estimates indicate that regions with generally high BCG vaccination rates have estimated COVID-19 deaths rates 10 to 14 times greater than reported estimates while regions with low BCG vaccination rates have estimated mortality rates 1.4 to 2 times greater than reported estimates [18], a bias that would tend to produce a spurious finding of vaccine efficacy [19]. Fourth, countries have, to varying extents, implemented a range of interventions to limit the spread of SARS-CoV-2 and ecologic studies to date have not been able to fully adjust for this variation. Fifth, previous studies have yet to standardise for different distributions of effect modifiers (e.g. age, sex, risk factors). Finally, analyses that are snapshots in time do not account for the dynamic nature of the pandemic as countries have reached peaks in COVID-19 burden at different times and have typically experienced a series of sub-epidemics within their borders.

Subsequent ecologic studies of the relationship between BCG vaccination and COVID-19 attempted to overcome these biases. For example, several studies attempted to account for potential exposure misclassification by using vaccine coverage rather than policy as the exposure of interest [11, 20–28]. Others used multivariable frameworks to account for confounding [11, 22, 23, 25, 27–35], but important confounding variables such as health care access, health care quality, and health consciousness were typically not addressed. Moreover, potential differential outcome misclassification bias, coexistence of COVID-19 mitigation policies, and the dynamically changing nature of the pandemic remain and have yet to be fully accounted for in these analyses.

## Conclusion

Although ecologic relationships may point the way for additional research, our findings suggest that significant caution is in order in their interpretation. We agree with the WHO's recommendation that, in the context of COVID-19, the BCG vaccine should only be used in RCTs. While we await those data, the present study underscores the need for caution in evaluating COVID-19 ecologic analyses, and highlights the limitations of this study design more broadly.

## Supporting information

**S1 File. Input data used for analyses.**
(XLSX)

## Author Contributions

**Conceptualization:** Peter Lurie, Mark N. Lurie.

**Data curation:** Jorge R. Ledesma, Rachel R. Yorlets.

**Formal analysis:** Jorge R. Ledesma, Rachel R. Yorlets, Stavroula Chrysanthopoulou.

**Investigation:** Jorge R. Ledesma, Peter Lurie.

**Methodology:** Jorge R. Ledesma, Peter Lurie, Stavroula Chrysanthopoulou, Mark N. Lurie.

**Project administration:** Garrison Daly.

**Supervision:** Mark N. Lurie.

**Visualization:** Jorge R. Ledesma, Stavroula Chrysanthopoulou.

**Writing – original draft:** Jorge R. Ledesma.

**Writing – review & editing:** Jorge R. Ledesma, Peter Lurie, Rachel R. Yorlets, Garrison Daly, Stavroula Chrysanthopoulou, Mark N. Lurie.

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
