## [Decision Letter · Decision Letter 0]

24 May 2022

PONE-D-21-36915Spurious early ecological association suggesting BCG vaccination effectiveness for COVID-19PLOS ONE

Dear Dr. Mark Lurie,

Thank you for submitting your manuscript to PLOS ONE. After careful consideration, we feel that it has merit but does not fully meet PLOS ONE’s publication criteria as it currently stands. Therefore, we invite you to submit a revised version of the manuscript that addresses the points raised during the review process.

Please submit your revised manuscript before Jul 03 2022 11:59PM. If you will need more time than this to complete your revisions, please reply to this message or contact the journal office at plosone@plos.org. Please include the following items when submitting your revised manuscript:A rebuttal letter that responds to each point raised by the academic editor and reviewer(s). You should upload this letter as a separate file labeled 'Response to Reviewers'.A marked-up copy of your manuscript that highlights changes made to the original version. You should upload this as a separate file labeled 'Revised Manuscript with Track Changes'.An unmarked version of your revised paper without tracked changes. You should upload this as a separate file labeled 'Manuscript'.

We look forward to receiving your revised manuscript.

Kind regards,

Wenping Gong, Ph.D.

Academic Editor

PLOS ONE

Journal Requirements:

ML and PL receive funding from the Sydney E. Frank Foundation

3. Thank you for stating the following in the Acknowledgments/Funding Section of your manuscript: 

This work was supported by the Sidney E Frank Foundation.

ML and PL receive funding from the Sydney E. Frank Foundation

Reviewers' comments:

Reviewer's Responses to Questions

**Comments to the Author**

1. Is the manuscript technically sound, and do the data support the conclusions?

Reviewer #1: Yes

Reviewer #2: Partly

2. Has the statistical analysis been performed appropriately and rigorously? 

Reviewer #1: Yes

Reviewer #2: No

3. Have the authors made all data underlying the findings in their manuscript fully available?

Reviewer #1: Yes

Reviewer #2: No

4. Is the manuscript presented in an intelligible fashion and written in standard English?

Reviewer #1: Yes

Reviewer #2: Yes

5. Review Comments to the Author

Reviewer #1: This is a useful contribution to the literature.

In the Conclusions, the authors frame the value of ecologic studies as hypothesis generating and findings should be interpreted with caution.

Ecologic studies certainly have value and indeed may be the most appropriate data for helping to answer certain questions (eg the impact of policy changes on population level outcomes). I would suggest moderating these claims.

Reviewer #2: Although authors referred to the Miller´s paper as the main point for the analysis, there are several other reports to be taken in consideration that authors omitted. Then a paper with only 14 references for such an important topic. It would recommended to work on those papers to enrich the discussion and conclusions.

6. PLOS authors have the option to publish the peer review history of their article (what does this mean?). If published, this will include your full peer review and any attached files.

Reviewer #1: No

Reviewer #2: **Yes: **Enrique Teran

---

## [Author Response · Author response to Decision Letter 0]

3 Jul 2022

Please see separate file "Response to reviewers"

---

## [Decision Letter · Decision Letter 1]

7 Sep 2022

Spurious early ecological association suggesting BCG vaccination effectiveness for COVID-19

PONE-D-21-36915R1

Dear Dr. Mark Lurie,

We’re pleased to inform you that your manuscript has been judged scientifically suitable for publication and will be formally accepted for publication once it meets all outstanding technical requirements.

Kind regards,

Wenping Gong, Ph.D.

Academic Editor

PLOS ONE

Additional Editor Comments (optional):

Reviewers' comments:

Reviewer's Responses to Questions

**Comments to the Author**

1. If the authors have adequately addressed your comments raised in a previous round of review and you feel that this manuscript is now acceptable for publication, you may indicate that here to bypass the “Comments to the Author” section, enter your conflict of interest statement in the “Confidential to Editor” section, and submit your "Accept" recommendation.

Reviewer #2: All comments have been addressed

2. Is the manuscript technically sound, and do the data support the conclusions?

Reviewer #2: Yes

3. Has the statistical analysis been performed appropriately and rigorously? 

Reviewer #2: N/A

4. Have the authors made all data underlying the findings in their manuscript fully available?

Reviewer #2: Yes

5. Is the manuscript presented in an intelligible fashion and written in standard English?

Reviewer #2: Yes

6. Review Comments to the Author

Reviewer #2: Thank you for take care of all suggestion given, particularly to increase the number of references. No further changes are necessary from my side.

7. PLOS authors have the option to publish the peer review history of their article (what does this mean?). If published, this will include your full peer review and any attached files.

Reviewer #2: **Yes: **Enrique Teran

---

## [Editor Report · Acceptance letter]

12 Sep 2022

PONE-D-21-36915R1 

Spurious early ecological association suggesting BCG vaccination effectiveness for COVID-19 

Dear Dr. Lurie:

I'm pleased to inform you that your manuscript has been deemed suitable for publication in PLOS ONE. Congratulations! Your manuscript is now with our production department. 

Kind regards, 

on behalf of

Dr. Wenping Gong 

Academic Editor

PLOS ONE